# The Smiles of Older People through Recreational Activities: Relationship between Smiles and Joy

**DOI:** 10.3390/ijerph18041600

**Published:** 2021-02-09

**Authors:** Ryuichi Ohta, Megumi Nishida, Nobuyasu Okuda, Chiaki Sano

**Affiliations:** 1Community Care, Unnan City Hospital, Unnan, Shimane 699-1221, Japan; 2Takenaka Corporation, Hommachi, Chuo-ku, Osaka 541-0053, Japan; nishida.megumi@takenaka.co.jp (M.N.); okuda.nobuyasu@takenaka.co.jp (N.O.); 3Department of Community Medicine Management, Faculty of Medicine, Shimane University, Izumo, Shimane 693-8501, Japan; sanochi@med.shimane-u.ac.jp

**Keywords:** elderly, joy, quality of smile, smile analysis application

## Abstract

Recreational activities are found to increase people’s smiles, arising joy in older people, but there is limited research on this topic within the Japanese context. This cross-sectional study aimed to measure the quality and frequency of smiles in older people living in rural settings using a smile analysis application. The participants comprised 13 females aged over 65 years who lived in Unnan City, Japan, and regularly attended recreational meetings. In this study, the recreational activity that the participants joined was a game called Mattoss. A video camera captured the participants’ faces, while a smile assessment application assessed their facial expressions for smiles and joy. A total of 2767 smiles were recorded. For validity, we calculated the Spearman’s rho score between smile and joy, which was 0.9697 (*p* < 0.001), while for reliability, we determined the Spearman’s rho score for each participant, which exceeded 0.7 (*p* < 0.001). Surges of smiles were induced by one’s own mistakes, successes, and big mistakes in the game and by severe or mild judgments by the referee. High validity and reliability of smile evaluation were demonstrated. The study found that smiling increased during recreational activities. Therefore, recreational activities can be encouraged for older people living in rural communities.

## 1. Introduction

Smiling can affect an individual’s physical and emotional well-being. It may induce various hormonal changes in the human body and improve several medical conditions [1,2] and has been related to quality of life (QOL) and enhanced health self-rating [3,4]. The quality and frequency of smiling is an essential factor in smile assessment [5]. The practical assessment of smiles can lead to more accurate estimates of people’s state of health. Previous studies have shown that people who frequently smile and laugh, live longer and healthier lives [5]. Although smiling can impact emotions in various ways, it has been shown that even artificially induced smiles can improve well-being [6,7]. However, it is uncertain whether the quality of a smile can lead to better health. Smiles’ features change from adolescence to old age due to the presence of wrinkles and a more relaxed muscle tone in older people; therefore smile quality should be assessed differently [8]. Maintaining health by smiling is essential, especially for older people, as their physical and mental abilities deteriorate due to the natural aging process [9]. To assess interventions designed to improve health and well-being, a tool for assessing older people’s smile quality should be developed.

Smile quality should be analyzed based on the meaning of facial expressions. Previous research suggests that the facial expression associated with a smile can result not only from the emotion of joy but also from amusement, embarrassment, nervousness, and politeness [7]. Among them, the emotion of joy can be vital to the enhancement of health; however, an artificial smile could also be effective [6]. There are limited studies focusing on older people and their smiles in the Japanese context. Therefore, our first research question was: “Are the facial expressions associated with smiles correlated with the emotion of joy?” 

Smiles can be triggered by recreational activities in communities. Since the Japanese society is one of the most rapidly aging [10], various recreational activities for older people are commonly organized to sustain the state of health of the elderly. Recreational activities can include physical and cognitive activities. There is significant evidence linking recreational activities such as exercise [11,12], gaming [13,14], and interaction with animals to health pursuits [15,16]. As recreational activities stimulate participants’ interests and induce joy, these activities can encourage healthy practices and smiles among participants [17], especially in rural areas where they are carried out in nature [18]. Triggered smiles can stimulate further interest and joy; however, few studies have examined the number of smiles and degree of joy during recreational activities, the relationship between them, and how smiles are triggered. Hence, our second research question was: “How frequently and by what stimuli are smiles triggered?”

In the United States, several tools have been developed to assess facial expressions, including Affdex, which has high validity and reliability for assessing smiles and joy [19]. However, no data from older people have been collected using this tool. Investigating the relationship between assessed smiles and joy among older people using this tool can facilitate its use in aging societies, thereby improving people’s overall health. In Japan and developed countries, the burden of the aging society, with its multimorbidity, on the health system can impact the well-being of the whole country [20,21]. Thus, creating conditions that induce smiling may have important and positive implications for the QOL of rural older people [22,23]. The purpose of this study was to assess if smiling increases in older people during recreational activities.

## 2. Materials and Methods

This study was conducted in the rural communities of Unnan City, Shimane, Japan, which borders the Hiroshima Prefecture to the South. In 2017, the total population of Unnan City was 38,882 (male = 18,720, female = 20,162), with 37.82% of the population aged 65 years and above. 

The 13 participants were females aged over 65 years who were living in Unnan City and who regularly participated in recreational meetings. They were recruited by Unnan City Hall at a community center. The present study was advertised in the local newspapers of the community. The recreational activities were organized by the community center’s staff and director. All participants joined the recreational activities voluntarily. The measurement of smiling facial expressions and the emotion of joy was conducted during the recreational activity.

The recreational activity consisted of a board game called Mattoss, containing three steps: physical exercise, point calculation, and communication. In preparation for the activity, all participants start with stretching their bodies—mainly arms and chest. Then, one participant throws four balls of different weights on a 100 cm^2^ board with 16 square spaces. Based on the position of the balls on the board, the participant calculates the score according to given rules and communicates the score to the next thrower among the participants, while cheering her/him on. Through this exercise, participants utilize their physical, mental, and cognitive functions (Appendix A). 

The participants’ data were collected at the community center. Data regarding their age and gender, previous participation in recreational meetings, and the number of family members with whom they lived were collected. Measurements of smiles (facial expression of smile) and joy were performed by the Automotive AI of Affdex, which was developed by Affectiva and customized into Japanse version by CAC corporation [19,24]. This application can detect four emotion metrics and eight facial expression metrics, including joy and smiles. Affdex emotion metrics have been trained and tested on complex datasets. In Affdex, smile was assessed based on the Affectiva database of facial expressions of smile, and joy was assessed by how the participants raised their cheeks and pulled up their lip corners in comparison to the baseline features of their faces [19,25]. To verify the validity and reliability of the tool, hundreds of thousands of facial frames from more than 3.2 million facial videos were used. The participants were from 87 countries, including Japan, and included individuals over 65 years old [8]. The Affdex can estimate the emotions of human beings with high accuracy (85.05%) [25]. This application analyzes human facial expressions at a rate of approximately 33 expressions per minute. 

By using this application, the facial expressions of the participants of this study could be measured while they performed recreational activities. Firstly, each participant’s face was registered by the Affdex program for facial recognition. Secondly, before starting the recreational activity, measurements were taken using a camera. The camera was located opposite to the participants while they were playing Mattoss, and could focus on all the participants’ faces collectively. The measurement of smiles and the game of Mattoss started simultaneously.

### Analysis

The values of the continuous variables are reported as the median (IQR) or mean (SD) according to the assessment of normality of the distribution by the Kolmogorov–Smirnov test. Smile and joy were measured as continuous variables. The levels of smile and joy were calculated using a score from 0 to 100 (0 = no smile and 100 = maximum smile or joy). For this study, a score of more than 80 was defined as a smile [25]. Background data are illustrated descriptively. To calculate how frequently the participants appeared, the number of frames in which each participant’s face appeared in the recording was determined. To assess validity and reliability, Spearman’s p was calculated to investigate the relationship between the scores for smiles and joy for each participant. Regarding sample size calculation, 12 participants were needed with 80% statistical power and 5% type 1 error to detect significant validity and reliability of 0.9. The significance level was set at *p* < 0.05, with a two-tailed assessment. The timing of the surges in smiles was analyzed by reviewing the video of the recreational activity from the perspective of the reasons for smiling. A surge was defined as a period of time with more than 10 smiles/minute and was established from the 75th quartile of the number of smiles/minute. In the analysis, the research team members (RO, MN, and NO) determined the graphical relationship of elapsed time and smile counts after viewing the video recording several times. Each researcher coded each smile surge. Afterwards, the researchers discussed each coding and clarified themes regarding reasons for smile surges. As an audit trial, the analysis was shown to the participants to verify whether the analysis accurately represented their real experiences.

The purpose and procedure of the study were explained to the participants, and informed consent was obtained. The Unnan City Hospital Ethics Committee approved this study, and the study is in accordance with the standards of the Declaration of Helsinki (Approval number: 20200015).

## 3. Results

All 13 participants of this study were women with an average age of 82.2 years (standard deviation = 5.2). All participants took part in recreational meetings. A total of 92.3% (12 out of 13) of the participants lived with their families. The total recorded time of the recreational activity was 45 min and 30 s, with a total number of 81,403 frames and 2767 smiles. Analysis of the scores related to the occurrences of smile and joy by the Kolmogorov–Smirnov test revealed absence of normal distribution, with a significance probability of *p* = 0.05 (Table 1). Spearman’s rho value was 0.969 (*p* < 0.001) between the scores for smiles and joy. Spearman’s rho value for each participant was higher than 0.7 (*p* < 0.001) (Table 2).

### 3.1. The Timing of Smile Surges and the Reasons behind Them

Based on the analysis of the graphical relationship between elapsed time and smile counts in the recorded video, three reasons underlying the surges of smiles appeared to be participants’ self-deprecation due to their own mistakes, successes and big mistakes in the game, and severe or mild judgment by the referee (Figure 1).

#### 3.1.1. Self-Deprecation Due to Own Mistakes—Timing: 06:40 (11 Smiles), 22:10 (10 Smiles)

One of the participants threw the ball, which crossed the border of the board. The participant described her mistake with the metaphor, “what goes around comes around,” in reference to the difficulties in her life. The other participants agreed with this statement; one of the participants said: “We are truly enjoying this game with no embarrassment. We can share our stress daily.” Since the participants lived in the same community, they openly talked of their life difficulties in their common context. Another participant stated: “Interesting! I can understand your experience. I’ve also experienced the same difficulties in my life.” Through the recreational activity, the participants had the opportunity to briefly reflect on usual events and released their stress.

#### 3.1.2. Successes and Big Mistakes in the Game—Timing: 06:45 (18 Smiles), 22:15 (14 Smiles), 35:10 (10 Smiles)

When the participants were unsuccessful and made big mistakes, other participants reacted positively to the situation, which stimulated smiles. One of the participants stated: “Good challenge! You are so nervous about your behavior. You must be a great thrower.” Another participant said: “You did well. You surely practiced a lot. You will be able to gain more points.” Their attitudes were focused on the results of the game, which created a relaxed environment and encouraged the participants to smile.

#### 3.1.3. Severe or Mild Judgment by the Referee—Timing: 22:40 (18 Smiles), 35:00 (10 Smiles), 35:05 (15 Smiles)

When the referee judged the ball positions and the participants’ mistakes severely or mildly, the participants cheerfully criticized the referee about the judgment. The participants interacted with the referee with cheerful criticism regarding the fact that the referee was strict. Their friendly interactions made the circumstances engaging and inclusive, which induced their smiles. One of the participants stated amicably: “Your judgment is too strict! Your judgment is not fair. Everybody thinks so.” Since the participants knew each other, including the referee, they became frank and open with one another and were able to release their stress. Another participant stated: “This exercise is for our relaxation. We can play more freely without thinking about criticism and enjoy the game regardless of the results.” The participants recognized how to enjoy the activity and perceived the referee’s judgments as humorous, which caused them joy.

## 4. Discussion

Smile assessment can be applied to diverse groups of people in the Japanese context. It can support health promotion among older people in rural contexts. Based on the present research, recreational activities which entail physical, mental, and cognitive stimulation can induce smiles in older people effectively. Based on previous studies, older populations, especially in rural areas, have difficulties interacting with others, which can negatively affect their health and increase depression and feelings of social isolation [20,26,27]. As smiling is an essential factor for the improvement of QOL, future studies should clarify the relationships between smiles and recreational activities, such as community activities that aim to keep older people active and motivated to live healthily [28]. The clarification of the quality of smiles in each community activity could justify the need for increased and improved recreational activities to enhance the QOL of older people. However, there are few studies investigating the effect of recreational activities on the number of smiles by people involved in them. Furthermore, consistent participation in recreational activities may be necessary for continuous health improvement in rural older people. Future studies should investigate the continual effect of recreational activities on rural older peoples’ smiles.

Recreational activities can trigger an increase in the number of smiles due to self-deprecation of one’s own mistakes, successes and big mistakes in a game, and severe or mild judgments by referees. As this research shows, people living in rural communities are familiar with each other, as they share the same social contexts. Because of this, recreational activities in rural communities can be effective and performed efficiently if the participants are not nervous and do not feel distant from one another [29,30]. On the other hand, in this study, there were variations of the frequency of smiles among the participants. This frequency could be affected by the location of the video camera; however, there could also be participants who did not experience joy from the recreational activity because they had become used to being alone and were struggling to connect with other participants [31]. As society becomes more diverse and various individuals become part of the same communities, recreational activities can be adjusted to the various needs of older people [32,33,34]. Social isolation is one of the significant challenges faced by rural communities [35,36]. In urban areas, there are various activities for social inclusion for isolated people. As a solution to this challenge, communities should host recreational activities and other events to trigger more smiles among older people and enhance their health and well-being [37,38]. In rural settings, there is an increased organization of recreational activities in combination with problem-solving social meetings, in which people discuss not only their health issues but also social problems with community health workers and local government officials [39]. It is recommended that future studies investigate the perceptions of rural people about recreational activities in their communities. Governments should implement flexible recreational activities in communities—facilitated by motivated older people—that can contribute to the solution of social problems such as isolation.

This study has four limitations. First, the sample size was restricted to a single small rural community. Therefore, the study may not be representative of a larger population, which caused selection bias. To overcome this limitation, we collected many samples of facial expressions from the participants to achieve high internal validity. Since we did not perform any multivariate analysis, such as logistic regression, we suggest that future studies investigate the relationship between the frequency of smiles and the participants’ backgrounds. Second, all participants were female. Nevertheless, the present results demonstrated that the application system is flexible and could be adjusted to recognize and evaluate aging-induced facial changes. Third, this study used one camera to record the participants’ facial expression, which limited the collection of facial expressions of the participants located in the periphery of the camera’s field of view. Future studies should use multiple cameras to improve data recording. Fourth, the participants were all from the same small rural community in Japan. Therefore, future studies should include more participants from other communities in Japan for better generalizability, because different personality traits, lifestyles, and socioeconomic status can affect the smiles of individuals. This investigation would ensure a high level of study generalizability.

## 5. Conclusions

The application-based smile assessment described in this study can be used with older populations, maintaining high validity and reliability. The smiles of older people increase during recreational activities including physical and mental exercises, which can depend on the closeness of relationships between people in a community. Therefore, recreational activities can be encouraged for older people in rural communities.

## Figures and Tables

**Figure 1 ijerph-18-01600-f001:**
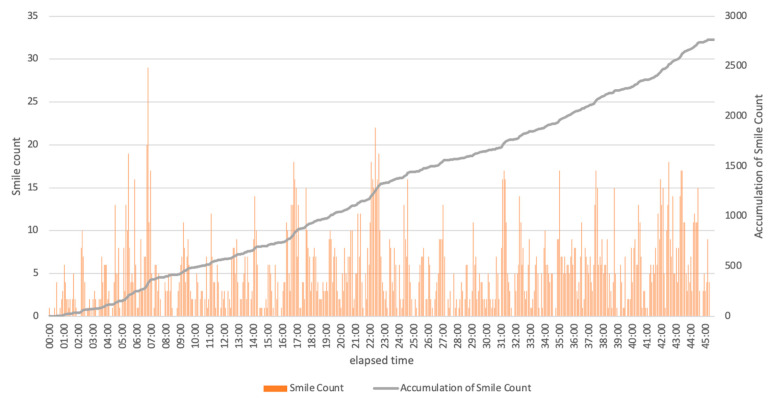
Relationship between elapsed time and smile counts.

**Table 1 ijerph-18-01600-t001:** Scores for smiles and joy.

Variable	Smiles			Joy		
	Median	Min	Max	Median	Min	Max
Participant 1	13.6	0.4	98.3	16.9	0.1	98.3
Participant 2	0.6	0.4	98.6	0.5	0.1	98.8
Participant 3	60.5	0.4	98.6	38.1	0.1	98.6
Participant 4	16.4	0.4	98.4	20.5	0.1	94.7
Participant 5	53.7	0.4	98.7	31.0	0.1	99.1
Participant 6	0.4	0.4	95.8	0.1	0.1	91.7
Participant 7	0.4	0.4	98.5	0.1	0.1	98.7
Participant 8	65.7	0.4	98.7	27.9	0.1	98.9
Participant 9	1.7	0.4	98.6	3.6	0.1	98.1
Participant 10	3.9	0.4	98.7	6.0	0.1	99.3
Participant 11	45.4	0.4	98.7	16.0	0.1	99.1
Participant 12	26.2	0.4	98.7	21.2	0.1	99.5
Participant 13	0.5	0.4	94.8	0.2	0.1	55.3

**Table 2 ijerph-18-01600-t002:** Correlation between smile and joy scores.

Variable	Number of Frames	Frequency of Appearance	Number of Smiles	Spearman’s p	*p*-Value
Total	81,403	6.213	2767	0.9697	<0.001
Participant 1	6418	0.078	30	0.9415	<0.001
Participant 2	45,346	0.554	40	0.933	<0.001
Participant 3	66,021	0.807	416	0.9632	<0.001
Participant 4	28,803	0.352	118	0.9157	<0.001
Participant 5	65,066	0.795	686	0.9822	<0.001
Participant 6	15,140	0.185	4	0.9639	<0.001
Participant 7	53,488	0.654	48	0.924	<0.001
Participant 8	32,309	0.395	330	0.9655	<0.001
Participant 9	65,258	0.798	215	0.9767	<0.001
Participant 10	61,479	0.751	167	0.9381	<0.001
Participant 11	70,440	0.861	486	0.954	<0.001
Participant 12	42,648	0.521	226	0.9633	<0.001
Participant 13	3258	0.040	1	0.7841	<0.001

## Data Availability

All relevant data sets in this study are described in the manuscript.

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
