# Peer review of "The Smiles of Older People through Recreational Activities: Relationship between Smiles and Joy"

_ijerph, 2021, doi:10.3390/ijerph18041600_

Round 1
Reviewer 1 Report
This manuscript offers a description of an innovative approach to assess the impact of interventions designed to enhance quality of life.
My primary critique is related to display of the results.
- Figure 2: needs a clear title, assumedly this data is from all 13 participants. The accumulated number of smiles over time would be assumed. The number of smiles at the onset of the count is not explained.
- items 3.1.1 to 3.1.3 - these sections explain the trigger for 81 smiles, yet thousand of smile were recorded. The rationale for addressing such a small portion of the total data is not clear.
Other critiques
- Research question is stated "Are the facial expressions of smiles correlated with emotions of joy". Yet the discussion does not relate the results to the research question. The outcome would be no, not correlated joy but with self-deprecations, extreme successes and mistakes, and/or judgement by referees?
- Is the sample size sufficient to validate an instrument?
Author Response
Reviewer 3
This manuscript offers a description of an innovative approach to assess the impact of interventions designed to enhance quality of life.
My primary critique is related to display of the results.
- Figure 2: needs a clear title, assumedly this data is from all 13 participants.The accumulated number of smiles over time would be assumed. The number of smiles at the onset of the count is not explained.
Response:
We thank the reviewer for this insightful comment. In response to this comment, we have revised the description by adding the onset of the count in the method section.
- items 3.1.1 to 3.1.3- these sections explain the trigger for 81 smiles, yet thousand of smile were recorded. The rationale for addressing such a small portion of the total data is not clear.
Response:
We thank the reviewer for this valuable comment. We have now added a description regarding the process of the qualitative analysis in the method section.
Other critiques
- Research question is stated "Are the facial expressions of smiles correlated with emotions of joy". Yet the discussion does not relate the results to the research question.The outcome would be no, not correlated joy but with self-deprecations, extreme successes and mistakes, and/or judgement by referees?
Response:
Thank you for pointing this out. In response to this comment, we have revised the purpose of the research.
- Is the sample size sufficient to validate an instrument?
Response:
Thank you for pointing this out. In response to this comment, we have added the description regarding the sample size calculation in the method section.
Reviewer 2 Report
The manuscript entitled “The smiles of older people through recreational activities: the relationship between smiles and joy” presents interesting issue, but some areas must be corrected.
Major:
Authors gathered some interesting data, but the version which they presented seems to be just the draft of their manuscript. It should be prepared based on proper scientific standard as the other scientific research in the journal. Authors should get familiar with the scope of various sections to address the specific needs of the readers.
Authors presented photos of the participants of their study, but without information about the informed consent of participants – it should be explained if they provided their consent for such publication.
It seems that there is a typo in the name of Author – is it Ohta or Ohtaa?
Abstract:
Lines 10-14 – Authors should reduce information associated with justification of the study – it should be presented, but it should not be excessive
Authors should present the detailed methodology of their study – what was planned within the recreational meetings? how were the smiles recorded? how was joy assessed?
Introduction:
It is not justified to present data for the period of COVID – as Authors stated that during the COVID pandemic older people are required to remain at home, the recreational activities presented in the conducted study do not solve the problem.
Materials and Methods:
Figure 1 – should be rather presented as a supplementary material, not in the main body of the study
Authors have mentioned correlation of the results with joy (line 94), but the question arises about validation of the association in Asian population, as well as in the population of elderly – it should be explained in what populations was the association observed, as the facial expression may differ between various populations.
Authors should present the detailed methodology of the presented study (what was done step by step) – Authors should present it with sufficient details to allow other researchers reproducing their study.
It seems that Authors did not verify the normality of distribution and they treated all the variables as normally distributed.
Authors should (1) verify the normality of distribution, (2) present their results - for normally distributed data present mean and SD values, but for the other distributions – present median, min and max values, (3) apply adequate statistical tests, that are based on the distribution.
Ethical considerations – number of decision (or date) should be presented.
Results:
Authors should apply adequate statistical tests, that are based on the distribution.
Authors should deepen their analysis – e.g. is the association influenced by any independent variable (Authors should include additional analysis)
The results presented so far (one simple table) is not sufficient to base a scientific paper on it.
Figure 2 – is not needed, as it does not present any results.
Lines 125-156 – should be rather presented as a supplementary material, as it presents rather description of the experiences, than the results of the study
Discussion:
Authors should be more focused on the “real” discussion – they should not reproduce in
Formation presented above or justification of the study should not be presented in this section.
Authors should in their discussion include 3 areas: (1) compare gathered data with the results by other authors, (2) formulate implications of the results of their study and studies by other authors, (3) formulate the future areas which should be studied
Author Contributions:
It seems that contribution of some Authors (NO, MN) was only minor and they did not participate in preparing manuscript (if not, it should be reflected in the manuscript). There is a serious risk of a guest authorship procedure which is forbidden. In such case they should be rather presented in Acknowledgements Section and not be indicated as authors of the study.
References:
Authors should include adequate references, while self-citations should be avoided, as they are not adequate (not associated with the aim of the study itself).
Author Response
Reviewer 1
The manuscript entitled “The smiles of older people through recreational activities: the relationship between smiles and joy” presents an interesting issue, but some areas must be corrected.
Major:
Authors gathered some interesting data, but the version which they presented seems to be just the draft of their manuscript. It should be prepared based on the same proper scientific standard as the other scientific researches in the journal. Authors should get familiar with the scope of various sections to address the specific needs of the readers.
Authors presented photos of the participants of their study, but without information about the informed consent of participants – it should be explained if they provided their consent for such publication.
It seems that there is a typo in the name of Author – is it Ohta or Ohtaa?
Response:
We thank the reviewer for this insightful comment. We agree with the suggestion. In response to this comment, we have revised typological errors intensively in the whole manuscript and added the issue about the informed consent of participants.
Abstract:
Lines 10-14 – Authors should reduce information associated with justification of the study – it should be presented, but it should not be excessive
Authors should present the detailed methodology of their study: What was planned within the recreational meetings? How were the smiles recorded? How was joy assessed?
Response:
Thank you for this comment. In response to this, we have revised the description of the abstract, reduced the introduction, added the kind of recreational activity, and how the smile and joy were assessed. Please see the revised abstract in the main text.
Introduction:
It is not justified to present data for the period of COVID – as Authors stated that during the COVID pandemic older people are required to remain at home, the recreational activities presented in the conducted study do not solve the problem.
Response:
Thank you for pointing this out. In response to this comment, we have deleted the description regarding COVID-19 and recreational activities.
Materials and Methods:
Figure 1 – should be rather presented as a supplementary material, not in the main body of the study.
Response:
Thank you for this suggestion. In response to this comment, we have deleted Figure 1 and moved it to supplementary material.
Authors have mentioned the correlation of the results with joy (line 94), but the question arises about validation of the association in Asian population, as well as in the population of elderly – it should be explained in what populations was the association observed, as the facial expression may differ between various populations.
Response:
We thank the reviewer for this insightful comment. We have revised the description of the application of Affidex accordingly regarding the validity with respect to Asian populations, gender, and age in the materials and methods section.
Authors should present the detailed methodology of the presented study (what was done step by step) – Authors should present it with sufficient details to allow other researchers to reproduce their study.
Response:
Thank you for this valuable suggestion. We have since revised the description of the methodology with sufficient details.
It seems that Authors did not verify the normality of distribution and they treated all the variables as normally distributed. Authors should (1) verify the normality of distribution, (2) present their results - for normally distributed data present mean and SD values, but for the other distributions – present median, min and max values, (3) apply adequate statistical tests, that are based on the distribution.
Ethical considerations – number of decision (or date) should be presented.
Response:
Thank you very much for your comment. We agree with the suggestion. We have since revised the description of the analysis section and results by adding the explanation of normality and additional analysis regarding each variable and including table 1. Additionally, we added the approval number of the ethical committee.
Results:
Authors should apply adequate statistical tests that are based on the distribution.
Response:
Thank you for pointing this out. In response to this comment, we have revised the analysis section and results by adding the explanation of normality and the additional analysis regarding each variable.
Authors should deepen their analysis – e.g. is the association influenced by any independent variable (Authors should include additional analysis). The results presented so far (one simple table) is not sufficient to base a scientific paper on it.
Response:
Thank you for your insightful comment. Our research was not able to conduct a logistic regression analysis using the background information of the participants as independent variables. Therefore, we have revised our limitations in the discussion by adding our limitations in the analysis and the possibility of future studies to investigate the relationship between the frequency of smiles and the participants’ backgrounds.
Figure 2 – is not needed, as it does not present any results.
Response:
We thank you for pointing this out. In response to this comment, we have revised the description of our qualitative analysis to show the need for the graphical representation of the smile surges.
Lines 125-156 – should be rather presented as a supplementary material, as it presents rather description of the experiences, than the results of the study
Response:
We thank the reviewer for this valuable comment. We have revised the methods and results sections to show that these are actually the results of the analysis of the contents of this research, together with the process of the analysis.
Discussion:
Authors should be more focused on the “real” discussion – they should not reproduce in Formation presented above or justification of the study should not be presented in this section.
Authors should in their discussion include 3 areas: (1) compare gathered data with the results by other authors, (2) formulate implications of the results of their study and studies by other authors, (3) formulate the future areas which should be studied
Response:
Thank you for this insightful comment. We have revised the discussion accordingly by adding the three suggested discussion areas. Please see the main file for the extensive revisions.
Author Contributions:
It seems that contribution of some authors (NO, MN) was only minor and they did not participate in preparing manuscript (if not, it should be reflected in the manuscript). There is a serious risk of a guest authorship procedure which is forbidden. In such case, they should be rather presented in Acknowledgements Section and should not be indicated as authors of the study.
Response:
Thank you for pointing this out. We have therefore revised the methods section to show their contribution to this study by adding their contribution to qualitative and quantitative analysis.
References:
Authors should include adequate references, while self-citations should be avoided as they are not adequate (not associated with the aim of the study itself).
Response:
We agree with the suggestion. We have since deleted our own references to avoid self-citations.
Reviewer 3 Report
Thank you for the opportunity to review this interesting manuscript. I have a few minor comments for the authors to consider:
- Abstract:
- Line 17 and 18: The symbol of rho score was incorrectly written as the letter ‘p’. Please correct it, or change “Spearman’s p score” to “Spearman’s rho score”
- Line 21: Please report the validity and reliability score to support the statement regarding ‘high’ validity and reliability.
- Line 21-22: “Smiling increases due to recreational activities and community engagement”. It would be good to report the statistical finding earlier, e.g. line 20, to support such statement.
- The last sentence needs to be revised because the results had yet shown that smile is directly related to increasing joy and QoL. They could be embarrassed (self-mistakes, extreme mistakes in games were noted as reasons of smiles by the authors), which may result in negative emotions, not necessarily just joy. In addition, the authors’ recommendation was based on only 13 elderly females – it may be good to tone it down or just highlight the potential instead of using ‘should be’ in the conclusion.
- Materials and Methods:
- The convenient sampling used in this study could lead to bias. Please discuss the implication of selection bias and limited generalisability of the findings in the discussion section.
- Please elaborate on the measures of smiles as well as joy. For example, certain angles of x degrees of the mouth and eyes are recognised as a smile (or joy?)
- Line 94: Please report the actual value of high correlation instead of just citing REF27.
- Line 101: Please justify the use of 80 as the cut-off point for smile, perhaps by citing a reference.
- Please clarify how joy was defined. Was there a cut-off point as well?
- Line 104: The symbol of rho score was incorrectly written as the letter ‘p’. Please correct it, or change “Spearman’s p score” to “Spearman’s rho score”
- Line 106: Did the authors mean p<0.05? If so, please correct the typing error.
- Line 108: Please cite a reference or provide a rationale to justify the definition of surge as more than 10 smiles, and clarify if this was 10 smiles per minute (or per frame?)
- Line 113: Please include the ethics approval number.
- The results indicated that the authors attempted to analyse the conversations qualitatively. As such, please include descriptions of the qualitative methods used in the results section.
- Results:
- Line 119, 120: The symbol of rho score was incorrectly written as the letter ‘p’. Please correct it, or change “Spearman’s p score” to “Spearman’s rho score”
- There appears to be some subjective opinions from the authors when reporting the qualitative results. It is fine if the descriptions of these opinions were the collective interpretation of the research team, which were then validated by the participants (which the authors could describe the process in the methods section), otherwise please remove the subjective opinions in the results section. The authors could discuss their opinions in the discussion section instead.
- Discussions:
- The first 3 sentences in the discussion section need to be deleted or revised. Please remember the whole study was only done on 13 people, and was a cross-sectional study, based on one type of game.
- The selection bias issue associated with convenient sampling method needs to be added alongside the limitation around small sample size. The findings need to be interpreted with cautions.
- The authors could also discuss if unknown personality traits or mood/events on the day affected the expression of smile. This is a typical limitation of a cross-sectional study.
- Please include the above limitations, and remove point 3 (line 201-202) as limitation. This sentence could be moved to the last sentence (line 205 onwards).
- Conclusions:
- Please revise (or delete) the last sentence as it is not currently related to the topic of smile and joy, nor the findings of this study.
Author Response
Thank you for the opportunity to review this interesting manuscript. I have a few minor comments for the authors to consider:
Abstract:
Line 17 and 18: The symbol of rho score was incorrectly written as the letter ‘p’. Please correct it, or change “Spearman’s p score” to “Spearman’s rho score”
Response:
Thank you for your valuable suggestion. We have now revised it to Spearman’s rho score.
Line 21: Please report the validity and reliability score to support the statement regarding ‘high’ validity and reliability. Line 21-22: “Smiling increases due to recreational activities and community engagement”. It would be good to report the statistical finding earlier, e.g. line 20, to support such statement.
Response:
We thank the reviewer for this insightful comment. In response to this comment, we have revised the description by adding the terms of validity and reliability.
The last sentence needs to be revised because the results had yet shown that smile is directly related to increasing joy and QoL. They could be embarrassed (self-mistakes, extreme mistakes in games were noted as reasons of smiles by the authors), which may result in negative emotions, not necessarily just joy. In addition, the authors’ recommendation was based on only 13 elderly females – it may be good to tone it down or just highlight the potential instead of using ‘should be’ in the conclusion.
Response:
Thank you for pointing this out to us. We have since revised the sentence to be more suggestive instead of definitive: “Therefore, recreational activities can be encouraged for older people in rural communities.”
Materials and Methods:
The convenient sampling used in this study could lead to bias. Please discuss the implication of selection bias and limited generalizability of the findings in the discussion section.
Response:
Thank you for this comment. In response to this, we have added mention of selection bias as a limitation to our study in our discussion:
“The sample size was restricted to a single small rural community, so only a small number of samples could be collected. Therefore, the study may not be representative of the larger population, leading to selection bias.”
Please elaborate on the measures of smiles as well as joy. For example, certain angles of x degrees of the mouth and eyes are recognized as a smile (or joy?)
Response:
Thank you for your valuable insight. We have since revised the description of the assessment of smile and joy in Affdex:
“In Affdex, smile was assessed based on the Affectiva database of facial expression of smile, and joy was assessed by how raised their cheeks and how pulled up their lip corners were compared to the baseline of the participants’ faces.”
Line 94: Please report the actual value of high correlation instead of just citing REF27.
Response:
We thank you for pointing this out. We have now revised the description of the assessment of Affdex by including the actual number.
Line 101: Please justify the use of 80 as the cut-off point for smile, perhaps by citing a reference.
Response:
We thank the reviewer for this insightful comment. In response to this comment, we have added a reference for the cut-off point.
Please clarify how joy was defined. Was there a cut-off point as well?
Response:
Thank you for your comment. We have since revised the description of the assessment of smile and joy in Affdex:
“In Affdex, smile was assessed based on the Affectiva database of facial expression of smile, and joy was assessed by how raised their cheeks and how pulled up their lip corners were compared to the baseline of the participants’ faces.” Besides, the cut-off point of the score of smile was 80 based on the previous studies.
Line 104: The symbol of rho score was incorrectly written as the letter ‘p’. Please correct it, or change “Spearman’s p score” to “Spearman’s rho score”
Response:
Thank you for your valuable suggestion. We have now revised it to Spearman’s rho score.
Line 106: Did the authors mean p<0.05? If so, please correct the typing error.
Response:
Thank you for pointing this out. We have revised the typological error accordingly.
Line 108: Please cite a reference or provide a rationale to justify the definition of surge as more than 10 smiles, and clarify if this was 10 smiles per minute (or per frame?)
Response:
We thank the reviewer for this insightful comment. In response to this, we have revised standard setting of smile surges in the methods section.
Line 113: Please include the ethics approval number.
Response:
Thank you for your comment. We have since added the approval number of the ethical committee.
The results indicated that the authors attempted to analyse the conversations qualitatively. As such, please include descriptions of the qualitative methods used in the results section.
Response:
We thank the reviewer for this valuable comment. We have now added a description regarding the process of the qualitative analysis in the method section.
Results:
Line 119, 120: The symbol of rho score was incorrectly written as the letter ‘p’. Please correct it, or change “Spearman’s p score” to “Spearman’s rho score”
Response:
Thank you for your valuable suggestion. We have now revised it to Spearman’s rho score.
There appears to be some subjective opinions from the authors when reporting the qualitative results. It is fine if the descriptions of these opinions were the collective interpretation of the research team, which were then validated by the participants (which the authors could describe the process in the methods section), otherwise please remove the subjective opinions in the results section. The authors could discuss their opinions in the discussion section instead.
Response:
We thank the reviewer for pointing this out. The description regarding the qualitative results have been revised accordingly and we have omitted the subjective sections.
Discussions:
The first 3 sentences in the discussion section need to be deleted or revised. Please remember the whole study was only done on 13 people, and was a cross-sectional study, based on one type of game.
Response:
We thank the reviewer for this insightful comment. We agree with the suggestion. We have since deleted the three aforementioned sentences.
The selection bias issue associated with convenient sampling method needs to be added alongside the limitation around small sample size. The findings need to be interpreted with cautions.
Response:
Thank you for this comment. In response to this, we have added mention of selection bias as a limitation to our study in our discussion:
“The sample size was restricted to a single small rural community, so only a small number of samples could be collected. Therefore, the study may not be representative of the larger population, leading to selection bias.”
The authors could also discuss if unknown personality traits or mood/events on the day affected the expression of smile. This is a typical limitation of a cross-sectional study.
Response:
Thank you for your valuable comment. We have now added a section in our discussion to address this:
“Therefore, future studies should include more participants from other countries because different personality traits, lifestyles, and socioeconomic status can affect the smiles of individuals.”
Please include the above limitations and remove point 3 (line 201-202) as limitation. This sentence could be moved to the last sentence (line 205 onwards).
Response:
We thank the reviewer for this insightful comment. We agree with the suggestion. We have revised limitations of our study by moving point 3 to the last part of the section.
Conclusions:
Please revise (or delete) the last sentence as it is not currently related to the topic of smile and joy, nor the findings of this study.
Response:
Thank you for pointing this out. We have since omitted the last sentence in the conclusion.
Round 2
Reviewer 1 Report
Thank you for considering my comments. Some additional clarification is needed to understand the application of this work
Abstract states: The study found that smiling increases due to recreational activities and community engagement.
Is the smiling part of feelings related to self-depreciation, extreme successes and mistakes, etc. and instead should be stated: smiling increases during recreational activities and community engagement
Introduction states: the purpose of the study is to assess if recreational activities increase smiles in/n older people. Confusing as here the community engagement has been dropped. Recreational activity and community engagement are not the same, are the terms being used interchangeably? As I read through the revised manuscript, I continue to be concerned that recreational activity is being used as proxy for community engagement. This distinction is important if this research will inform interventions/programs with older people. Do programs need to provide opportunities for recreational activity and community engagement. Having facilities for solo recreational activities is quite different than offering gatherings. Would solo recreational activities yield an increase in smiling?
The conclusion is based on an underlying assumption that recreational activities trigger smiles, perhaps it is not the recreational activity but the community engagement.
Table 1: I do not understand the "out of order numbers" on the far left side or throughout the table.
Author Response
Comment:
Thank you for considering my comments. Some additional clarification is needed to understand the application of this work
Abstract states: The study found that smiling increases due to recreational activities and community engagement.
Is the smiling part of feelings related to self-depreciation, extreme successes and mistakes, etc. and instead should be stated: smiling increases during recreational activities and community engagement?
Response:
We thank the reviewer for this insightful comment. We agree with the suggestion. In response to this comment, we have revised the abstract by adding the suggested sentence.
Comment:
Introduction states: the purpose of the study is to assess if recreational activities increase smiles in/n older people. Confusing as here the community engagement has been dropped. Recreational activity and community engagement are not the same, are the terms being used interchangeably? As I read through the revised manuscript, I continue to be concerned that recreational activity is being used as proxy for community engagement. This distinction is important if this research will inform interventions/programs with older people. Do programs need to provide opportunities for recreational activity and community engagement. Having facilities for solo recreational activities is quite different than offering gatherings. Would solo recreational activities yield an increase in smiling?
Response:
We thank the reviewer for this insightful comment. We agree with the suggestion. In response to this comment, we have revised the background by deleting the word “community engagement” and emphasizing the assessment of single recreational activity.
The conclusion is based on an underlying assumption that recreational activities trigger smiles; perhaps, it is not the recreational activity but the community engagement.
Comment:
Table 1: I do not understand the "out of order numbers" on the far left side or throughout the table.
Response:
We thank the reviewer for this insightful comment. We agree with the suggestion. In response to this comment, we have revised Table 1 by deleting the numbers.
Reviewer 2 Report
The manuscript entitled “The smiles of older people through recreational activities: the relationship between smiles and joy” presents interesting issue, but some areas must be corrected.
Results:
Authors should deepen their analysis – e.g. is the association influenced by any independent variable (Authors should include additional analysis)
Figure 2 – is not needed, as it does not present any results.
Lines 160-191 – should be rather presented as a supplementary material, as it presents rather description of the experiences, than the results of the study
Author Contributions:
It seems that contribution of some Authors (NO, MN) was only minor and they did not participate in preparing manuscript (if not, it should be reflected in the manuscript). There is a serious risk of a guest authorship procedure which is forbidden. In such case they should be rather presented in Acknowledgements Section and not be indicated as authors of the study. If they participated in preparing manuscript, it should be clearly stated.
Author Response
Comment:
The manuscript entitled “The smiles of older people through recreational activities: the relationship between smiles and joy” presents interesting issue, but some areas must be corrected.
Results:
Authors should deepen their analysis – e.g. is the association influenced by any independent variable (Authors should include additional analysis)
Response:
We thank the reviewer for this insightful comment. We agree with the suggestion. However, we did not collect more data except for the data in this article, so we added the description for next studies studies, including the collection of other independent variables.
Comment:
Figure 2 – is not needed, as it does not present any results.
Lines 160-191 – should be rather presented as a supplementary material, as it presents rather description of the experiences, than the results of the study
Response:
We thank the reviewer for this insightful comment. We agree with the suggestion. However, we believe that this qualitative analysis support small sample analysis in the quantitative analysis, so we would like to retain Figure 1 and the qualitative analysis.
Comment:
Author Contributions:
It seems that contribution of some Authors (NO, MN) was only minor and they did not participate in preparing manuscript (if not, it should be reflected in the manuscript). There is a serious risk of a guest authorship procedure which is forbidden. In such case they should be rather presented in Acknowledgements Section and not be indicated as authors of the study. If they participated in preparing manuscript, it should be clearly stated.
Response:
We thank the reviewer for this insightful comment. We agree with the suggestion. In response to this comment, we have revised the role descriptions in the author contributions section.